# Diagnostic Yield of Cardiac Magnetic Resonance in Athletes with and without Features of the Athlete’s Heart and Suspected Structural Heart Disease

**DOI:** 10.3390/ijerph19084829

**Published:** 2022-04-15

**Authors:** Łukasz A. Małek, Barbara Miłosz-Wieczorek, Magdalena Marczak

**Affiliations:** 1Department of Epidemiology, Cardiovascular Disease Prevention and Health Promotion, National Institute of Cardiology, 04-635 Warsaw, Poland; 2Magnetic Resonance Unit, Department of Radiology, National Institute of Cardiology, 04-635 Warsaw, Poland; barbara-milosz@tlen.pl (B.M.-W.); mmarczak@ikard.pl (M.M.)

**Keywords:** differential diagnosis, late gadolinium enhancement, cardiomyopathy, myocarditis, myocardial infarction

## Abstract

Cardiac magnetic resonance (CMR) is a second-line imaging test in cardiology. Balanced enlargement of heart chambers called athlete’s heart (AH) is a part of physiological adaptation to regular physical activity. The aim of this study was to evaluate the diagnostic utility of CMR in athletes with suspected structural heart disease (SHD) and to analyse the relation between the coexistence of AH and SHD. We wanted to assess whether the presence of AH phenotype could be considered as a sign of a healthy heart less prone to development of SHD. This retrospective, single centre study included 154 consecutive athletes (57 non-amateur, all sports categories, 87% male, mean age 34 ± 12 years) referred for CMR because of suspected SHD. The suspicion was based on existing guidelines including electrocardiographic and/or echocardiographic changes suggestive of abnormality but without a formal diagnosis. CMR permitted establishment of a new diagnosis in 66 patients (42%). The main diagnoses included myocardial fibrosis typical for prior myocarditis (*n* = 21), hypertrophic cardiomyopathy (*n* = 17, including 6 apical forms), other cardiomyopathies (*n* = 10) and prior myocardial infarction (*n* = 6). Athlete’s heart was diagnosed in 59 athletes (38%). The presence of pathologic late gadolinium enhancement (LGE) was found in 41 patients (27%) and was not higher in athletes without AH (32% vs. 19%, *p* = 0.08). Junction-point LGE was more prevalent in patients with AH phenotype (22% vs. 9%, *p* = 0.02). Patients without AH were not more likely to be diagnosed with SHD than those with AH (49% vs. 32%, *p* = 0.05). Based on the results of CMR and other tests, three patients (2%) were referred for ICD implantation for the primary prevention of sudden cardiac death with one patient experiencing adequate intervention during follow-up. The inclusion of CMR into the diagnostic process leads to a new diagnosis in many athletes with suspicion of SHD and equivocal routine tests. Athletes with AH pattern are equally likely to be diagnosed with SHD in comparison to those without AH phenotype. This shows that the development of AH and SHD can occur in parallel, which makes differential diagnosis in this group of patients more challenging.

## 1. Introduction

Cardiac magnetic resonance (CMR) is one of the second-line, non-invasive imaging tests in cardiology [1,2]. According to the global cardiovascular magnetic resonance registry, the main clinical indications for CMR include analysis of cardiomyopathies (21%) followed by assessment of myocardial viability or stress CMR in chest pain syndrome (both 16%) and evaluation of etiology of arrhythmias or planning of electrophysiological studies (15%) [3]. Due to well-balanced spatial and temporal resolution, lack of exposition to radiation or problems with the imaging window and the possibility of visualizing myocardial scars with means of late gadolinium enhancement (LGE), this method is particularly useful in the analysis of athletes with suspected structural heart disease [4]. Initial tests performed in athletes with suspected cardiovascular conditions such as electrocardiogram (ECG), echocardiography, Holter ECG or exercise test are often inconclusive due to the limited diagnostic potential of these methods and difficulties in differentiation between the physiological adaptation of the heart to exercise and pathological changes [5,6]. This is because athletes may present features of the so-called athlete’s heart (AH) including balanced enlargement of most heart chambers accompanied sometimes by mild left ventricular (LV) wall thickening, and borderline systolic and supra-normal diastolic function of the LV [7,8]. All of these adaptive changes may occasionally blur the diagnostic picture. However, differentiation between physiological adaptation to exercise and cardiac pathology is of paramount importance in terms of continued participation in competitive and/or intensive sport [6]. CMR has been shown to be the gold standard in the assessment of heart chamber size, myocardial function and mass [9]. Nevertheless, it is still underused as a diagnostic method in athletes, which may be partially attributed to the low availability of trained personnel, costs of the test and knowledge on the indications for testing [10]. In cardiomyopathies, which are the most common indication for CMR, the referral rate according to the European registry was only 29.4% and varied largely across the centres (1–63%) [11]. For all of these reasons, it is difficult to assess the diagnostic yield of this method in athletes.

Therefore, we decided to perform a retrospective, single centre analysis of consecutive CMR tests performed in athletes referred for a scan due to suspected structural heart disease (SHD) based on symptoms and initial tests. We wanted to analyze how the new findings available with CMR compare to the effectiveness of CMR in published data from the general population. Additionally, we decided to compare the diagnostic yield in athletes with and without the AH phenotype. We wanted to assess whether the presence of AH phenotype could be considered as a sign of a healthy heart less prone to development of SHD.

## 2. Materials and Methods

### 2.1. Study Group

This retrospective analysis included consecutive athletes admitted to the Sports Cardiology Ambulatory Clinic in the National Institute of Cardiology in Warsaw, Poland, between July 2019 and December 2021 who were referred for CMR due to suspected SHD, but without a formal diagnosis. This is one of the main tertiary centers admitting athletes from the Mazovian region, but also from other parts of Poland. Indications for CMR followed current statements and recommendations in sports cardiology [5,6]. Suspicion of SHD was based on resting ECG/Holter ECG and echocardiographic findings suggestive of abnormality with or without symptoms. The most common symptoms and abnormalities observed in initial tests leading to CMR are presented in Table 1.

Because of the COVID-19 pandemic, we excluded athletes referred for CMR testing due to suspected SARS-CoV-2 infection or vaccination complications such as acute myocarditis, pericarditis or acute myocardial infarction occurring up to 3 months post COVID-19 or vaccination. Furthermore, our group has already published the analysis of routine CMR testing in athletes with a positive COVID-19 test [12]. The athletes spanned many sport disciplines including running, triathlon, cycling, pentathlon, rowing, skating (endurance), football, basketball, handball, cross-fit (mixed), weightlifting, boxing (power) and fencing (skill).

### 2.2. Definitions

Athletes were defined as elite if they competed on a national or international level and trained over 10 h a week, semi-professional/professional if they competed at the regional level and trained at least 6 h a week, and amateur if they trained less, but not fewer than 3 hours/week and/or did not engage regularly in competitions [6]. Sports disciplines were divided into endurance, power, mixed and skill based on the published criteria [13]. Athlete’s heart was defined as a balanced enlargement of most heart chambers above the reference values in adults with or without mild left ventricular hypertrophy [7,8]. Structural heart diseases were diagnosed based on the current recommendations [14]. Acute myocarditis was diagnosed according to the updated Lake Louise criteria using a T2-based criterion in combination with a T1-based criterion [15].

### 2.3. CMR Protocol and Analysis

MR imaging was performed with a Siemens Magnetom Avanto Fit 1.5 Tesla scanner (Siemens, Erlangen, Germany). The protocol included initial scout images, followed by cine balanced steady-state free precession (bSSFP) breath-hold sequences in 2-, 3-, and 4-chamber views. Short axis was identified using the 2- and 4-chamber images and was followed by the acquisition of a stack of short-axis images, which included the ventricles from the mitral and tricuspid valvular plane to the apex. Pre-contrast T1-mapping with modified Look Locker sequence (MOLLI) and T2-mapping were performed with a T2- prepared SSFP sequence immediately after acquisition of the bSSFP cine images when diagnostically necessary and were processed using MyoMaps software (Siemens, Erlangen, Germany). For that purpose, three short-axis slices (one basal, one mid-ventricular and one apical) and 2-, 3-, and 4-chamber views were obtained. Following these acquisitions, 0.1 mmol/kg of a gadolinium contrast agent (gadobutrol–Gadovist^®^, Bayer Shering Pharma AG, Berlin, Germany) was administered and flushed with 30 mL of isotonic saline. Late gadolinium enhancement (LGE) images in three long-axis and a stack of short-axis imaging planes were obtained with a breath-hold phase-sensitive inversion recovery sequence (PSIR) 10 min after the contrast injection. The inversion time was adjusted to null normal myocardium (typically between 250 and 350 ms as assessed using a TI-scout acquisition). In cases of ischemia analysis, hyperemia was obtained with means of 400 mg of i.v. regadenoson injection (Haupt Pharma, Wolfratshausen, Germany) with a first-pass stress and/or rest perfusion. Additionally, breath-hold phase contrast velocity mapping was performed in the ascending aorta (at the level of the sinotubular junction) and the main pulmonary artery (located at the midpoint of the blood vessel). Velocity encoding sensitivity was adjusted to avoid aliasing.

Images were analysed with the use of dedicated software (Syngovia, Siemens, Erlangen, Germany). All studies were assessed independently by three physicians—one cardiologist and two radiologists, each with long-lasting expertise in CMR (Ł.A.M.—14 years of experience, B.M.-W. and M.M.—13 years of experience). End-diastolic and end-systolic endocardial and epicardial contours were drawn semi-automatically for the left ventricle (LV) and manually for the right ventricle (RV) in the short axis stack of bSSFP cine acquisitions. Delineated contours were used for the quantification of end-diastolic (LVEDVI/RVEDVI) and end-systolic volumes (LVESVI/RVESVI), ejection fraction (LVEF/RVEF), and LV mass (LVMI), indexed to body surface area. We used previously published normal values of left and right ventricular volumes, systolic function and mass as a reference [16].

The presence and location of LGE were assessed visually. Junction point (insertion/hinge point) LGE in isolation was not considered as pathologic [4]. Abnormal native T1 and T2 values were defined as greater than 1054 ms and greater than 50 ms, respectively, based on previously derived sequence and scanner-specific cut-offs of 2 SDs above the respective means in a healthy population [17].

### 2.4. Clinical Follow-Up 

Clinical follow-up included referral for ICD implantation and analysis of adequate device interventions in the study period. 

### 2.5. Statistical Analysis

All results for categorical variables are presented as a number and percentage. Continuous variables are expressed as the median and interquartile range (IQR) or mean and standard deviation (SD) depending on the normality of distribution assessed with means of the Kolmogorov–Smirnov test. Either the chi-square test or the Fisher exact test was used for the comparison of categorical variables, when appropriate. Student’s t-test or Mann–Whitney test for unpaired samples was applied to compare two continuous variables depending on the data distribution. All tests were two-sided with a significance level of *p* < 0.05. Statistical analyses were performed with MedCalc statistical software 10.0.2.0 (Ostend, Belgium).

## 3. Results

### 3.1. Baseline Characteristics

Of the 421 athletes admitted to the Clinic in the study period due to suspected heart disease and after exclusion of athletes studied for suspected complications of SARS-CoV-2 infection or vaccination, a total of 154 athletes were included (36% of the whole group). 

The mean age of athletes referred for CMR was 34 ± 12 years and 87% of them were male. The study group included 39 elite (25%), 19 semi-professional or professional (13%) and 96 amateur (62%) athletes. Most of them practiced endurance disciplines (*n* = 102, 67%), followed by mixed sports (*n* = 36, 23%), power (*n* = 14, 9%) and skill sports (*n* = 2, 1%).

### 3.2. CMR Findings

CMR permitted establishment of a new diagnosis in 66 patients (42%). The main diagnoses included myocardial fibrosis typical for prior myocarditis (*n* = 21), hypertrophic cardiomyopathy (HCM—*n* = 17, including 6 apical forms), other cardiomyopathies (*n* = 10) and prior myocardial infarction (MI, *n* = 6). The presence of pathologic LGE was found in 41 patients (27%). The examples of athletes who were diagnosed with SHD are presented in Figure 1.

### 3.3. Athlete’s Heart and CMR Result

Athlete’s heart was found in 59 athletes (38%) in the studied group. Examples of athletes with and without features of an AH are presented in Figure 2. Patients with an AH phenotype were more likely found in the elite group (42% vs. 15%, *p* = 0.0003) as demonstrated in Table 2. The prevalence of pathologic LGE was not higher in athletes without AH in comparison to those with AH phenotype (32% vs. 19%, *p* = 0.08). Junction-point LGE was more prevalent in patients with AH phenotype (22% vs. 9%, *p* = 0.02). Patients without AH were not more likely diagnosed with SHD than those with AH (49% vs. 32%, *p* = 0.05).

### 3.4. Clinical Follow-Up

Based on the results of CMR and other tests, three patients (2%) were referred for ICD implantation for the primary prevention of sudden cardiac death (one case of dilated cardiomyopathy—DCM and two cases of arrhythmogenic cardiomyopathy—AC). The patient with DCM experienced an adequate ICD intervention in the study period.

## 4. Discussion

We have shown that with the use of CMR it is possible to confirm or make a new diagnosis of structural heart disease in over 40% of athletes with equivocal results of initial testing. A formal diagnosis helps to guide further management in this group of patients in line with recently updated ESC recommendations in sports cardiology [6]. It also shortens the time of uncertainty for the athlete often related to periods of mandated competitive sport cessation and involuntary detraining. This is crucial, especially for professional athletes, with regard to their return to play and continued professional career. Similarly high prognostic potential of CMR in a real-life clinical settings has been demonstrated in data from EuroCMR registry, which was a multi-centre initiative with consecutive enrolment of over 27,000 general-population patients from 57 centres in 15 countries [18]. In 61.8% of cases, CMR findings impacted patient management and in nearly 8.7% of patients the final diagnosis based on CMR was different from the initial one, leading therefore to complete change in management. Similarly high diagnostic potential of CMR, now demonstrated also in athletes, is related to the detection of subtle or less visible transthoracic echocardiography pathological changes. It is particularly important in athletes, where the presence of borderline features of cardiomyopathies is more likely than in the general population as more advanced forms of diseases usually lead to earlier diagnosis and elimination from intensive or professional sport [6]. Many diseases are also caught at an early stage of development where the full clinical picture may not be present yet as many athletes are young. Examples of such borderline or less severe disease phenotypes include less pronounced forms of HCM with a larger predilection for apical HCM, dilated cardiomyopathy (DCM) with only mildly reduced systolic function or arrhythmogenic cardiomyopathy (AC) with subtle regions of akinesia/dyskinesia of the RV without markedly decreased global systolic function [19,20,21].

Another group of findings where CMR is particularly useful includes detection of small scars in the myocardium in cases where myocardial wall thickness or systolic function are not compromised. These small areas of fibrosis may arise from prior myocarditis, myocardial infarction or accompany even discrete forms of cardiomyopathies. Despite lack of influence on cardiac systolic performance such myocardial scars may be a substrate for potentially life-threatening arrhythmias, which require continuous monitoring [22,23]. The presence of LGE in our study was found in 27% of athletes, but it should be noted that our group included only athletes with suspicion of SHD. Meta-analyses of the prevalence of LGE, performed mainly in endurance athletes, demonstrated that LGE might be observed in over 21% of athletes and is more likely than in the control population [24]. However, many of these LGE, as in our study, were located in the junction point arising probably from increased tension in that area during prolonged hours of volume and pressure overload and therefore forming a part of the adaptive changes without documented impact on prognosis [4]. For this reason we decided not to include junction point LGE as pathologic in the current analysis.

It is important to note that CMR is free from ionizing radiation, and considered safe in terms of rare contrast administration complications. It also does not impact sports performance in any way, which is crucial for young and otherwise healthy athletes. Despite these advantages CMR is still underused in sports cardiology. A survey performed by D’Ascenzi et al. on the use of cardiac imaging in the evaluation of athletes in clinical practice including responses from 97 countries showed that CMR is used always or often after echocardiography in only 44% of symptomatic athletes and in only 6% of asymptomatic athletes. Among the barriers related to CMR highlighted by the respondents was low access to equipment, low coverage of screening costs by social/health insurance or lack of personnel training. Other mentioned barriers included also long waiting lists and lack of referral by other physicians [10]. Some of the barriers could have been overcome by providing and ascertaining guideline-based training in CMR [25]. We hope that the current work will serve as an argument towards a higher use of CMR in the testing of athletes. In our opinion, only centers equipped with easily accessible CMR can provide the full spectrum of diagnostics for athletes suspected of having a SHD. Although the cost of a single study may seem relatively high, it can translate into information otherwise available in several other diagnostic tests such as detailed cardiac function, morphology, tissue structure or functional testing. It may also obviate the need for close monitoring in pure AH cases.

Finally, we have demonstrated that pathological CMR findings, including LGE and features of SHD, are equally likely found in patients with and without AH phenotype. The development of AH features is considered as a healthy, physiological adaptation of the heart to regular physical activity. One could imagine that it is more likely to find SHD in athletes who do not present with AH phenotype, as their hearts may be less prone to physiological adaptation due to harboured disease. Lack of such a relation may be explained by the nature of factors leading to the most commonly found cardiovasacular diseases such as external conditions (viral infection) or mild congenital/idiopathic factors, where normal heart adaptation is not affected [14,26]. In our opinion, these findings are important, as they demonstrate that SHD can be superimposed on normal AH features, therefore, blurring the clinical picture and making the differential diagnosis more problematic. Features of AH should not be taken into consideration as a mitigating factor when SHD is suspected.

Our study has some limitations. First of all it was performed in one centre with a single referring point for CMR, which may constitute an inclusion bias. However, this is an example of a real-life clinical situation in a tertiary cardiologic centre dedicated to sports cardiology and all of the indications for CMR following published guidelines and statements. Secondly, due to the short time from the CMR studies, we were not able to collect other clinical follow-up data to analyse if CMR diagnoses impacted the prognosis of studied athletes, which might have further strengthened our results. We realize that our group presents a limited scope in terms of general practice in cardiology. However, there is a growing number of physically active people, mostly amateurs and not only professional athletes, who may also require the differentiation between AH and SHD. We believe that our study can serve as an example of how CMR can be used to increase the benefits of physical activity by clearance of athletes for return to play and, at the same time, to reduce the risk for athletes diagnosed with SHD based on the results of the study. Therefore, CMR has the potential to improve the risk–benefit ratio of physical activity.

## 5. Conclusions

The inclusion of CMR into the diagnostic process leads to a new diagnosis in many athletes with suspicion of SHD and equivocal routine tests. Athletes with AH pattern are equally likely to be diagnosed with SHD in comparison to those without AH phenotype. This shows that the development of AH and SHD can occur in parallel, which makes differential diagnosis in this group of patients more challenging.

## Figures and Tables

**Figure 1 ijerph-19-04829-f001:**
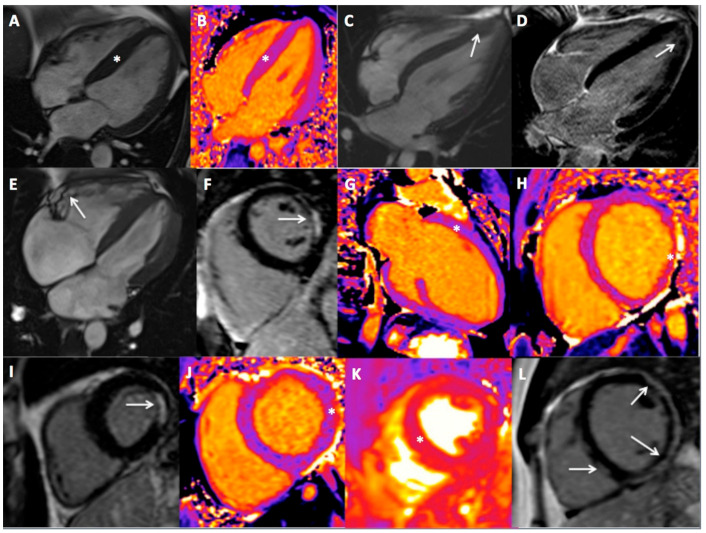
Examples of main cardiac magnetic resonance findings in the studied group. (**A**) 4-chamber cine view in an amateur triathlete with hypertrophic cardiomyopathy (HCM, asterisk), (**B**) 4-chamber T1-mapping view in an amateur footballer with HCM and visible increase of T1 time in the inter-ventricular septum (asterisk), (**C**,**D**) 4-chamber cine and LGE views in an professional footballer with apical HCM (arrow, C) and small areas of LGE (arrow, D), (**E**,**F**) 4-chamber cine and short axis LGE views in a semi-professional triathlete showing unbalanced enlargement of the right ventricle with areas of dyskinesia (arrow, E) accompanied by non-ischemic LGE in the left ventricle (arrow, F), (**G**) 2-chamber T1-mapping view in a professional footballer with dilated cardiomyopathy (DCM) and elevated T1 time (asterisk), (**H**) Short-axis T1-mapping view in an amateur runner with DCM and elevated T1-time (asterisk), (**I**) Short axis LGE view in an amateur veteran runner showing small ischemic scar post silent myocardial infarction (arrow), (**J**,**K**) Short axis T1-mapping and T2-mapping views in a professional volleyball player with acute myocarditis (elevated T1 and T2 time shown with asterisks), (**L**) Short axis LGE view in an amateur runner with prior myocarditis and extensive sub-epicardial LGE in the lateral wall and mid-wall LGE in the inter-ventricular septum (arrows).

**Figure 2 ijerph-19-04829-f002:**
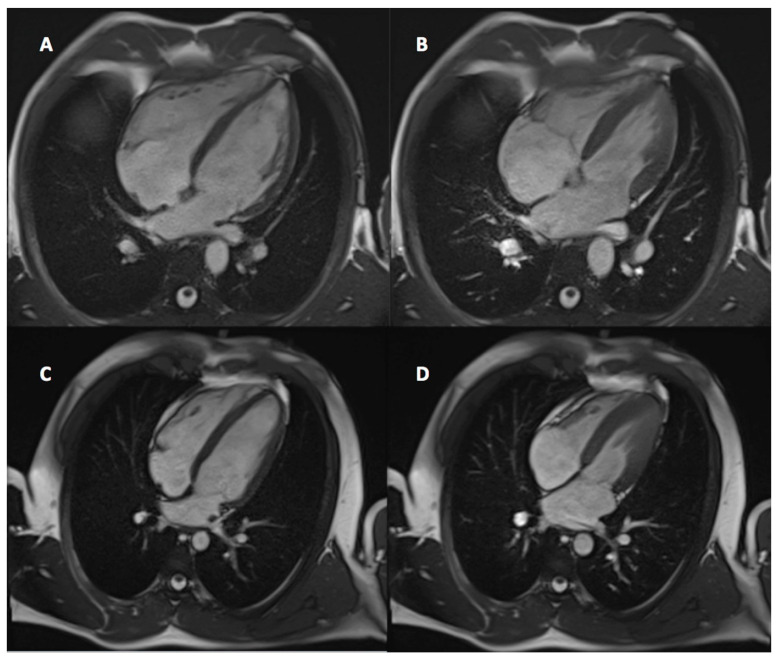
4-chamber view. Examples of a patient with athlete’s heart (AH) features ((**A**)—end-diastole, (**B**)—end-systole) and without AH features ((**C**)—end-diastole, (**D**)—end-systole).

**Table 1 ijerph-19-04829-t001:** The most common symptoms and abnormalities on initial testing leading to CMR.

Symptoms	Resting ECG/Holter ECG	Echocardiography
Chest pain	T-wave inversion in anterior, lateral or inferior leads	Left ventricular hypertrophy ≥13 mm
Palpitations	Premature ventricular contractions	Isolated left ventricular enlargement
Irregular heart beat	Non-sustained ventricular tachycardia	Isolated right ventricular enlargement
Loss of consciousness	Supraventricular arrhythmia	Decreased left ventricular ejection fraction
Reduced physical performance	LBBB or RBBB with axis deviation	Decreased right ventricular ejection fraction
Upper respiratory tract infection/fever	Pauses	Left ventricular hypertrabeculation

ECG—electrocardiogram, LBBB—left bundle branch block, RBBB—right bundle branch block.

**Table 2 ijerph-19-04829-t002:** Baseline characteristics and cardiac magnetic resonance (CMR) results in patients with and without athlete’s heart phenotype.

Parameter	Athlete’s Heart *n* = 59 (38%)	No Athlete’s Heart *n* = 95 (62%)	*p*-Value
**Baseline characteristics**			
Age (yrs, SD)	32 ± 13	35 ± 12	0.19
Male sex (*n*, %)	52 (88)	82 (86)	0.74
Athlete category (*n*,%)			<0.0001
amateur	25 (42)	71 (74)	0.0001
semi-or professional	9 (15)	10 (11)	0.54
elite	25 (42)	14 (15)	0.0003
Sport discipline (*n*, %)			0.45
endurance	40 (68)	62 (65)	0.88
mixed	15 (25)	21 (22)	0.78
power	4 (7)	10 (11)	0.57
skill	0 (0)	2 (2)	0.52
**CMR parameters**			
LVEDVI (mL/m^2^, SD)	115 ± 13	89 ± 13	<0.001
LVESVI (mL/m^2^)	47 ± 13	33 ± 8	<0.001
LVEF (%)	61 ± 7	63 ± 6	0.10
LVMI (g/m^2^)	85 ± 14	73 ± 16	<0.001
RVEDVI (mL/m^2^)	118 ± 15	93 ± 17	<0.001
RVESVI (mL/m^2^)	52 ± 12	39 ± 13	<0.001
RVEF (%)	58 ± 6	59 ± 7	0.35
LAA (cm^2^)	29 ± 6	24 ± 5	<0.001
RAA (cm^2^)	29 ± 6	25 ± 5	<0.001
IVSd (mm)	10.7 ± 1.4	10.9 ± 2.4	0.74
LGE in junction point	13 (22)	9 (9)	0.03
LGE other than junction point (*n*,%)	11 (19)	30 (32)	0.09
ischemic	1 (2)	5 (5)	0.41
non-ischemic	10 (17)	25 (27)	0.25
**CMR result**			
Disease (*n*,%)	19 (32)	47 (49)	0.05
Type of disease (*n*, %)			
HCM	1 (2)	10 (11)	0.05
HCM apical	2 (3)	4 (4)	1.00
All HCM	3 (5)	14 (15)	0.22
DCM	3 (5)	3 (3)	0.68
AC	1 (2)	3 (3)	1.00
LVNC	0 (0)	0 (0)	-
All cardiomyopathy	7 (12)	20 (21)	0.21
Prior MI	0 (0)	6 (6)	0.08
Acute/prior myocarditis	7 (12)	14 (14)	0.79
Other findings *	5 (8)	7 (7)	0.95

AC—arrhythmogenic cardiomyopathy, DCM—dilated cardiomyopathy, HCM—hypetrophic cardiomyopathy, IVSd—interventricular septal diameter, LAA—left atrial area, LGE—late gadolinium enhancement, LVEDVI—left ventricular end-diastolic volume index, LVEF—left ventricular ejection fraction, LVESVI—left ventricular end-systolic volume index, LVMI—left ventricular mass index, LVNC—left ventricular non-compaction, LVSVI—left ventricular stroke volume index, MI—myocardial infarction, RAA—right atrial area, RVEDVI—right ventricular end-diastolic volume index, RVEF—right ventricular ejection fraction, RVESVI—right ventricular end-systolic volume index. ***** dilated ascending aorta with tricuspid aortic valve (*n* = 5), biscuspid aortic valve without complications (*n* = 2), pericardial cyst (*n* = 1), anomalous origin of coronary artery with ischemia (*n* = 1), multiple left ventricular crypts (*n* = 1), mitral valve prolapse with regurgitation (*n* = 2).

## Data Availability

Data are available on request from the authors.

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
