# Peer review of "Diagnostic Yield of Cardiac Magnetic Resonance in Athletes with and without Features of the Athlete’s Heart and Suspected Structural Heart Disease"

_ijerph, 2022, doi:10.3390/ijerph19084829_

Round 1

Reviewer 1 Report

This is an interesting although highly specific paper on CMR use as a diagnostic tool for SHD.

In the Introduction it would be useful to have some additional information on the effectiveness of CMR use in the general population and the referral rates if available.

In the methodology section it would be helpful to:

a) to have some information on the coverage of the study centre and on the presence of other similar centres in the country to get an estimate of coverage

b) have some additional information on the specific criteria for exclusion of study sample observations due to post vaccination effects.

In the Discussion section, the authors refer to training of professionals in CMR. It would be interesting to see some specific suggestions on the specific requirements for training, as well as a proposal of the suggested criteria for the health insurance providers to compensate this examination . Finally, what could be the incentives for a specialised centre to acquire the specific equipment?

Author Response

This is an interesting although highly specific paper on CMR use as a diagnostic tool for SHD. In the Introduction it would be useful to have some additional information on the effectiveness of CMR use in the general population and the referral rates if available.

Thank you for this insightful remark. We have added the following information to the introduction section related to the use of CMR in general population and referral rates: „According to the global cardiovascular magnetic resonance registry the main clinical indications for CMR include analysis of cardiomyopathies (21%) followed by assessment of myocardial viability or stress CMR in chest pain syndrome (bot 16%) and evaluation of etiology of arrhythmias or planning of electrophysiological studies (15%) [3].” … „In cardiomyopathies, which are the most common indication for CMR the referral rate according to European registry was only 29.4% and varied largely across the centres (1-63%) [11].”The effectiveness of CMR in the general population is commented in the discussion section with the following sentences: „Similarly high prognostic potential of CMR in a real-life clinical settings has been demonstrated on a data from EuroCMR registry, which was a multi-centre initiative with consecutive enrolment of over 27000 patients from 57 centres in 15 countries [18]. In 61.8% of cases, CMR findings impacted on patient management and in nearly 8.7% of patients the final diagnosis based on CMR was different from the initial one, leading therefore to complete change in management.”

In the methodology section it would be helpful to:

  1. a) to have some information on the coverage of the study centre and on the presence of other similar centres in the country to get an estimate of coverage

            This is the only tertiary centre in Poland dedicated to Sports Cardiology equipped with on-hand CMR. Patients are referred mainly from the Masovian region, but also from other parts of Poland. Other centres either have Sports Cardiology Units or CMR facilities, but rarely both, so this is a unique infrastructure in Poland, but not in Europe. Therefore we are able to present a true estimate of the use of CMR in athletes with suspected cardiac conditions requiring CMR as most of them are referred to our centre. We have added the following sentence to the methodology section: „This is one of the main tertiary centers admitting athletes from the Mazovian region, but also from other parts of Poland.”

  1. b) have some additional information on the specific criteria for exclusion of study sample observations due to post vaccination effects.

We have modified the sentence in the Methodology section to include more specific criteria: “Because of the Covid-19 pandemic we have decided to exclude athletes referred for CMR testing due to suspected SARS-CoV-2 infection or vaccination complications such as acute myocarditis, pericarditis or acute myocardial infarction occurring up to 3 months post Covid-19 or vaccination.”

In the Discussion section, the authors refer to training of professionals in CMR. It would be interesting to see some specific suggestions on the specific requirements for training, as well as a proposal of the suggested criteria for the health insurance providers to compensate this examination. Finally, what could be the incentives for a specialised centre to acquire the specific equipment?

Thank you for this comment. We have provided additional information on the training guidelines in CMR: „Some of the barriers could have been overcome by providing and ascertaining guideline-based training in CMR [25].” We have also provided earlier a rationale for wider use of CMR in athletes: “It also shortens the time of uncertainty for the athlete often related to periods of mandated competitive sport cessation and involuntary detraining. This is crucial, especially for professional athletes, with regard to their return to play and continued professional career.” Finally, we have added the following sentences related to advantages of CMR for healthcare providers to compensate for the study or equipment: „In our opinion only centers equipped with easily accessible CMR can provide the full spectrum of diagnostics for athletes suspected of having a SHD. Although the cost of a single study may seem relatively high, it can translate into information otherwise available in several other diagnostic tests such as detailed cardiac function, morphology, tissue structure or functional testing. It may also obviate the need for close monitoring in pure AH cases.”

Reviewer 2 Report

This study is expected to serve as an important basis for presenting an advanced method of Cardiac Magnetic Resonance for the diagnosis of athlete's heart and structural heart disease. Please add content related to supplementation of the limited scope in the discussion.

Author Response

This study is expected to serve as an important basis for presenting an advanced method of Cardiac Magnetic Resonance for the diagnosis of athlete's heart and structural heart disease. Please add content related to supplementation of the limited scope in the discussion.

           Thank you for this comment. We realize that our group presents a limited scope in terms of general practice in cardiology. However, there is a growing number of physically active people, mostly amateurs and not only professional athletes, who may also require the differentiation between AH and SHD. We believe that our study can serve as an example how CMR can be used to increase the benefits of physical activity by clearance of athletes for return to play and, at the same time, to reduce the risk for athletes diagnosed with SHD based on the results of the study. Therefore, CMR has a potential to improve the risk-benefit ratio of physical activity, at least in equivocal cases. We have added this information to the discussion section: “We realize that our group presents a limited scope in terms of general practice in cardiology. However, there is a growing number of physically active people, mostly amateurs and not only professional athletes, who may also require the differentiation between AH and SHD. We believe that our study can serve as an example how CMR can be used to increase the benefits of physical activity by clearance of athletes for return to play and, at the same time, to reduce the risk for athletes diagnosed with SHD based on the results of the study. Therefore, CMR has a potential to improve the risk-benefit ratio of physical activity, at least in equivocal cases. We have added this information to the discussion section.”

Reviewer 3 Report

The manuscript “Diagnostic yield of cardiac magnetic resonance in athletes with and without features of the ahtlete’s heart and suspected structural heart disease” aimed to investigate the benefits of performing CMR in athletes that have indications for such. I commend the author for their job in putting the study together. General comments and specific points and sections are provided below:

General comments

Presentation: Tables and figures are well designed, and I commend the authors for that. English writing requires professional revisions.

Novelty: The observations are novel.

Title: The title is adequate.

Abstract: the abstract is concise, but a bit confusing. The rationale for the study is not clearly stated, not is the research question. Results should be further described within the abstract, also.

Introduction: The introduction is brief and should explain the rationale of the study better. What is the research question/problem? What were the a priori hypotheses? Was this an exploratory study? The whole study should be better justified in the introduction by improving the rationale.

Materials and Method: Methods are adequately described and sound. I missed the statement regarding ethical approval. Was the study approved by an institutional ethics committee prior to data assessments?

Results: The results are well described. Please refrain from presenting non-significant results as “higher” or “trending”. It is not reasonable to discuss non-significant changes as significant if the level of significance was not reached.

Discussion: The discussion is adequate, but based in non-significant changes in some parts.

Conclusion: The conclusion is adequate.

Please find specific comments detailed below:

L17: all sport categories?

L23: please provide the full wording for the acronym LGE within the abstract.

L45-46: this sentence does not connect with the previous sentence.

L63: consecutive athletes? Why not just athletes?

Author Response

The manuscript “Diagnostic yield of cardiac magnetic resonance in athletes with and without features of the ahtlete’s heart and suspected structural heart disease” aimed to investigate the benefits of performing CMR in athletes that have indications for such. I commend the author for their job in putting the study together. General comments and specific points and sections are provided below:

General comments

Presentation: Tables and figures are well designed, and I commend the authors for that. English writing requires professional revisions.

The manuscript has been read by a native speaker who introduced several corrections.

Novelty: The observations are novel.

Title: The title is adequate.

Abstract: the abstract is concise, but a bit confusing. The rationale for the study is not clearly stated, not is the research question. Results should be further described within the abstract, also.

Thank you for this comment. We have modified the abstract including more precise presentation of the rationale for the study, research question and more results: „The aim of the study was to evaluate the diagnostic utility of CMR in athletes with suspected structural heart disease (SHD) and to analyse the relation between the coexistence of AH and SHD. We wanted to assess whether the presence of AH phenotype could be considered as sign of a healthy heart less prone to development of SHD.” … „CMR permitted to establish a new diagnosis in 66 patients (42%). The main diagnoses included myocardial fibrosis typical for prior myocarditis (n=21), hypertrophic cardiomyopathy (n=17, including 6 apical forms), other cardiomyopathies (n=10) and prior myocardial infarction (n=6). Athlete’s heart was diagnosed in 59 athletes (38%). The presence of pathologic late gadolinium enhancement (LGE) was found in 41 patients (27%) and was not higher in athletes without AH (32% vs. 19%, p=0.08). Junction-point LGE was more prevalent in patients with AH phenotype (22% vs. 9%, p=0.02). Patients without AH were not more likely diagnosed with an SHD than those with AH (49% vs. 32%, p=0.05). Based on the results of CMR and other tests 3 patients (2%) have been referred for an ICD implantation in the primary prevention of sudden cardiac death with one patient experiencing adequate intervention during follow-up.”

Introduction: The introduction is brief and should explain the rationale of the study better. What is the research question/problem? What were the a priori hypotheses? Was this an exploratory study? The whole study should be better justified in the introduction by improving the rationale.

Thank you for this comment. We have modified the Introduction section to include the information on the use of CMR the general population and low frequency of referrals for CMR even in most common indications: “According to the global cardiovascular magnetic resonance registry the main clinical indications for CMR include analysis of cardiomyopathies (21%) followed by assessment of myocardial viability or stress CMR in chest pain syndrome (bot 16%) and evaluation of etiology of arrhythmias or planning of electrophysiological studies (15%) [3]….In cardiomyopathies, which are the most common indication for CMR the referral rate according to European registry was only 29.4% and varied largely across the centres (1-63%) [11].”, which should further justify the rationale of the study. We have also modified the paragraph dedicated to explanation of the aim of the study: „Therefore, we have decided to perform a retrospective, one-centre analysis of consecutive CMR tests performed in athletes referred for the scan due to suspected structural heart disease (SHD) as based on symptoms and initial tests. We wanted to analyze how the new findings available with CMR compare to the effectiveness of CMR in published data from the general population. Additionally, we have decided to compare the diagnostic yield in athletes with and without the AH phenotype. We wanted to assess whether the presence of AH phenotype could be considered as a sign of a healthy heart less prone to development of SHD.” We have also added the following statement to the Discussion section: „We realize that our group presents a limited scope in terms of general practice in cardiology. However, there is a growing number of physically active people, mostly amateurs and not only professional athletes, who may also require the differentiation between AH and SHD. We believe that our study can serve as an example how CMR can be used to increase the benefits of physical activity by clearance of athletes for return to play and, at the same time, to reduce the risk for athletes diagnosed with SHD based on the results of the study. Therefore, CMR has a potential to improve the risk-benefit ratio of physical activity, at least in equivocal cases. We have added this information to the discussion section.”

Materials and Method: Methods are adequately described and sound. I missed the statement regarding ethical approval. Was the study approved by an institutional ethics committee prior to data assessments?

This information is provided at the end of the manuscript as per journal guidelines: The study was conducted in accordance with the Declaration of Helsinki, and approved by the Institutional Ethics Committee of National Institute of Cardiology for the retrospective analysis of data (protocol code IK.NPIA.0021.22.1962/22, date of approval 10/02/2022). The approval was done after data assessments, which in our institution is a general practice for retrospective analyses.

Results: The results are well described. Please refrain from presenting non-significant results as “higher” or “trending”. It is not reasonable to discuss non-significant changes as significant if the level of significance was not reached.

            We have removed the statements “higher” or “trending” from all results with p>0.05. 

Discussion: The discussion is adequate, but based in non-significant changes in some parts.

         Thank you for this comment. We have removed the following part of the discussion focused on non-significant changes: “However, with the exception of HCM known of genetically determined muscle disarray and small myocardial infarctions, more likely occurring in less trained athletes [12,23], the other types of SHD were observed with similar prevalence irrespective of AH phenotype”

Conclusion: The conclusion is adequate.

Please find specific comments detailed below:

L17: all sport categories?

The athletes belonged to all sports categories (endurance, power, mixed and skill) as explained in the methods section. However following your comment we have grouped the disciplines by category in the Methodology section: “The athletes spanned over many sport disciplines including running, triathlon, cycling, pentathlon, rowing, skating (endurance), football, basketball, handball, cross-fit (mixed), weightlifting, boxing (power) and fencing (skill).”

L23: please provide the full wording for the acronym LGE within the abstract.

We have explained the abbreviation in the abstract.

L45-46: this sentence does not connect with the previous sentence.

We have modified these two sentences to sound as following: „It is because athletes may present features of the so-called athlete’s heart (AH) including balanced enlargement of most heart chambers accompanied sometimes by mild left ventricular (LV) wall thickening, borderline systolic and supra-normal diastolic function of the LV [6,7].”

L63: consecutive athletes? Why not just athletes?

We have used this term to specify that the group was not preselected in anyway.

Round 2

Reviewer 3 Report

Thank you for the revisions. I have no further comments.